# DIFFERENCE-SEEKING GENERATIVE ADVERSARIAL NETWORK

## ABSTRACT

We propose a novel algorithm, **D**ifference-**S**eeking **G**enerative **A**dversarial **N**etwork (DSGAN), developed from traditional GAN. DSGAN considers the scenario that the training samples of target distribution, $p_t$, are difficult to collect. Suppose there are two distributions $p_{\bar{d}}$ and $p_d$ such that the density of the target distribution can be the differences between the densities of $p_{\bar{d}}$ and $p_d$. We show how to learn the target distribution $p_t$ only via samples from $p_d$ and $p_{\bar{d}}$ (relatively easy to obtain). DSGAN has the flexibility to produce samples from various target distributions (*e.g.* the out-of-distribution). Two key applications, semi-supervised learning and adversarial training, are taken as examples to validate the effectiveness of DSGAN. We also provide theoretical analyses about the convergence of DSGAN.

## 1 INTRODUCTION

In machine learning, how to learn a probability distribution is usually conducted in a unsupervised learning manner. Generative approaches are developed for learning data distribution from its samples and thereafter produce novel and high-dimensional samples from learned distributions, such as image and speech synthesis (Saito et al. (2018)). The state-of-the-art approaches is so-called Generative Adversarial Networks (GAN)(Goodfellow et al. (2014)). GAN produces sharp images based on a game-theoretic framework, but can be tricky and unstable to train due to multiple interacting losses. Specifically, GAN consists of two functions: generator and discriminator. Both functions are represented as parameterized neural networks. The discriminator network is trained to classify whether or not inputs belong to real data or fake data created by the generator. The generator learns to map a sample from a latent space to some distribution to increase the classification errors of discriminator. GAN corresponds to a minimax two-player game, which ends if the generator actually learns the real data distribution. The generator is of main interest because the discriminator will be unable to differentiate between both distributions once the generator has been trained well.

### 1.1 MOTIVATIONS

In reality, it is difficult to collect training samples from unseen classes, which none of their samples involves the training phase, but their samples could be encountered in the testing phase. How to reject or recognize unseen data as "abnormal" (not belonging to the training data) is an important issue known as one-class classification (Ruff et al. (2018)). Due to the absence of unseen data, most of algorithms are unsupervised (Scholkopf & Smola (2001)). Based on the assumption that there do exist very few unseen examples, some approaches (Wu & Ye (2009)) focus on supervised learning using unbalanced data. In addition to one-class classification, Dai et al. (2017) theoretically show that complementary data, which is also considered as unseen data, can improve semi-supervised learning. Another related issues is adversarial attack, where classifiers may be vulnerable to adversarial examples, which are unseen during the training phase.

Fig. (8) in Appendix A illustrates the applications regarding unseen data. Apparently, if we can generate unseen data via GAN, it is helpful for those applications. But, traditional GAN requires preparing plenty of training samples of unseen classes for training, leading to the contradiction with the prerequisite. This fact motivates us to design the proposed DSGAN, which can generate unseen data by taking seen data as training samples shown in Fig. (1). The nuclear idea is to consider the distribution of unseen data as the difference of two distributions, which both are relatively easy to obtain. For example, out-of-distribution examples in the MNIST dataset, from another point of view, are found to belong to the difference of the set of examples in MNIST and the universal set.

Examples in both sets are relatively easy to obtain. It should be noted that the target distribution is equal to the training data distribution in traditional GANs; nevertheless, both distributions are considered different in DSGAN.

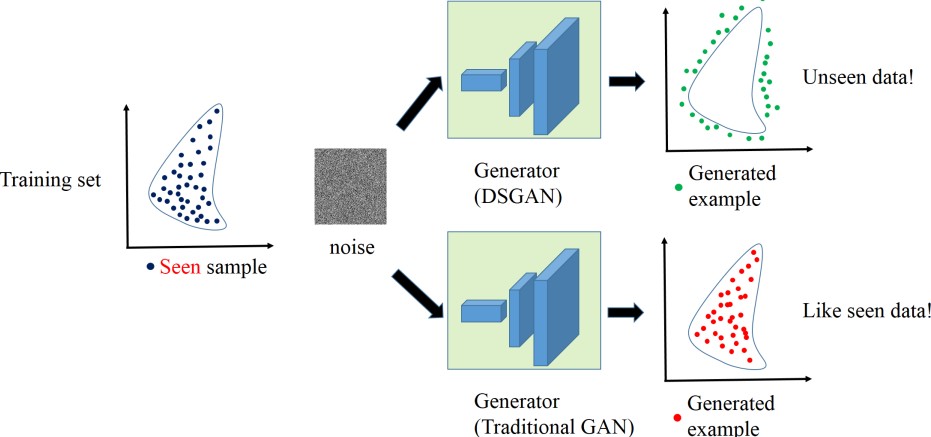

Figure 1: The illustration of difference between traditional GAN and DSGAN.

## 1.2 CONTRIBUTIONS

In this paper, we make the following contributions:

(1) We propose a novel algorithm, **D**ifference-**S**eeking **G**enerative **A**dversarial **N**etwork (DS-GAN), where the density of target distribution $p_t$ is the difference between those of any two distributions, $p_{\bar{d}}$ and $p_d$. Nevertheless, the differences of two densities being negative are not well-defined. Thus, instead of learning the target distribution $p_t$ directly, the generator distribution approximates $p_t$ by minimizing the statistical distance between the mixture distribution of $p_d$ and the generator distribution, and $p_{\bar{d}}$.

(2) Theoretical results based on traditional GAN are extended to the case of mixture distribution considered in this paper. With enough capacity of the generator and the discriminator, we show DSGAN can learn the generator distribution $p_g$ under mild condition where the support set of $p_g$ is the difference of support sets of $p_{\bar{d}}$ and $p_d$.

(3) We show that DSGAN possesses the flexibility to learn different target distributions in two key applications: semi-supervised learning and adversarial training. Samples from target distribution in semi-supervised learning must satisfy two conditions: i) are linear combination of any label data and unlabel data; ii) do not belong to neither label data nor unlabel data. For adversarial training, samples from the target distribution are assigned as out-of-distribution examples with bounded distortion. Experiments validate that DSGAN can learn these two kinds of distributions well in various datasets.

The paper is structured as follows. In Sec. 2, we introduce DSGAN, including the algorithm in the training procedure. Theoretical results are described in Sec. 3. In Sec. 4, two applications are taken as examples to show the effectiveness of DSGAN. Finally, we present the experimental results in Sec. 5 and conclude by pointing out some promising directions for future work in Sec. 7.

## 2 PROPOSED METHOD-DSGAN

### 2.1 THE SCHEME OF DSGAN

We denote the generator distribution as $p_g$ and training data distribution as $p_d$, both in a $N$-dimensional space. Let $p_{\bar{d}}$ be the distribution decided by user. For example, $p_{\bar{d}}$ can be the convolution of $p_d$ and normal distribution. Let $p_t$ be the target distribution which the user is interested in, and it can be expressed as

$$(1 - \alpha)p_t(x) + \alpha p_d(x) = p_{\bar{d}}(x), \tag{1}$$

where $\alpha \in [0, 1]$. Our method, DSGAN, aims to learn $p_g$ such that $p_g = p_t$. Intuitively, our method tries to learn $p_g$ such that $p_g(x) \sim \dfrac{p_{\bar{d}}(x) - \alpha p_d(x)}{1 - \alpha}$. In other words, the generator is going to output samples located in high-density areas of $p_{\bar{d}}$ and low-density areas of $p_d$.

At first, we formulate the generator and discriminator in GANs. The inputs $z$ of the generator are drawn from $p_z(z)$ in an $M$-dimensional space. The generator function $G(z; \theta_g) : \mathbb{R}^M \to \mathbb{R}^N$ represents a mapping to data space, where $G$ is a differentiable function with parameters $\theta_g$. The discriminator is defined as $D(x; \theta_d) : \mathbb{R}^N \to [0, 1]$ that outputs a single scalar. $D(x)$ can be considered as the probability that $x$ belongs to a class of real data.

Similar to traditional GAN, we train $D$ to distinguish the real data and the fake data sampled from $G$. Meanwhile, $G$ is trained to produce realistic data as possible to mislead $D$. But, in DSGAN, the definitions of "real data" and "fake data" are different from those in traditional GAN. The samples from $p_{\bar{d}}$ are considered as real but those from the mixture distribution between $p_d$ and $p_g$ are considered as fake. The objective function is defined as follows:

$$\min_G \max_D V(G, D) = \mathbb{E}_{x \sim p_{\bar{d}}(x)}\left[\log D(x)\right] + (1 - \alpha)\mathbb{E}_{z \sim p_z(z)}\left[\log\left(1 - D\left(G(z)\right)\right)\right]$$
$$+ \alpha\mathbb{E}_{x \sim p_d(x)}\left[\log\left(1 - D(x)\right)\right]. \tag{2}$$

During the training procedure, an iterative approach like traditional GAN is to alternate between $k$ steps of training $D$ and one step of training $G$. In practice, minibatch stochastic gradient descent via back propagation is used to update $\theta_d$ and $\theta_g$. In other words, for each of $p_g$, $p_d$ and $p_{\bar{d}}$, $m$ sample are required for computing gradients, where $m$ is the number of samples in a minibatch. Algorithm 1 illustrates the training procedure in detail. DSGAN suffers from the same drawbacks with traditional GAN (*e.g.*, mode collapse, overfitting, and strong discriminator such that the generator gradient vanishes). There are literatures (Salimans et al. (2016), Arjovsky & Bottou (2017), and Miyato et al. (2018)) focusing on improving the above problems, and their ideas can be combined into DSGAN.

Li et al. (2017) and Reed et al. (2016) proposed the similar objective function like (2). Their goal is to learn the conditional distribution of training data. Nevertheless, we aim to learn the target distribution $p_t$ in Eq. (1), not the training data distribution.

---

**Algorithm 1** The training procedure of DSGAN using minibatch stochastic gradient descent. $k$ is the number of steps to apply to the discriminator. $\alpha$ is the ratio between $p_g$ and $p_d$ in the mixture distribution. We used $k = 1$ and $\alpha = 0.8$ in experiments.

01.    **for** number of training iterations **do**
02.      **for** $k$ steps **do**
03.        Sample minibatch of $m$ noise samples $z^{(1)}, ..., z^{(m)}$ from $p_g(z)$.
04.        Sample minibatch of $m$ samples $x_d^{(1)}, ..., x_d^{(m)}$ from $p_d(x)$.
05.        Sample minibatch of $m$ samples $x_{\bar{d}}^{(1)}, ..., x_{\bar{d}}^{(m)}$ from $p_{\bar{d}}(x)$.
06.        Update the discriminator by ascending its stochastic gradient:

$$\nabla_{\theta_d}\left[\frac{1}{m}\sum_{i=1}^{m}\log D\left(x_d^{(i)}\right) + \log\left(1 - D\left(G\left(z^{(i)}\right)\right)\right) \right.$$
$$\left. + \log\left(1 - D\left(x_{\bar{d}}^{(i)}\right)\right)\right]$$

07.      **end for**
08.      Sample minibatch of $m$ noise samples $z^{(1)}, ..., z^{(m)}$ from $p_g(z)$.
09.      Update the generator by descending its stochastic gradient:

$$\nabla_{\theta_g}\frac{1}{m}\sum_{i=1}^{m}\left[\log\left(1 - D\left(G\left(z^{(i)}\right)\right)\right)\right]$$

10.    **end for**

---

## 2.2 CASE STUDY ON SYNTHETIC DATA AND MNIST

To get more intuitive understanding about DSGAN, we conduct several case studies on 2D synthetic datasets and MNIST. Those results validate that DSGAN can learn the distribution we desire.

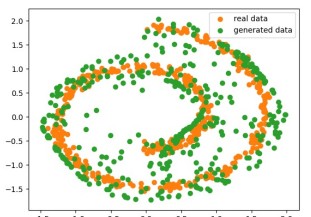
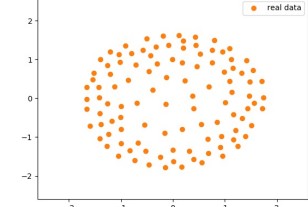
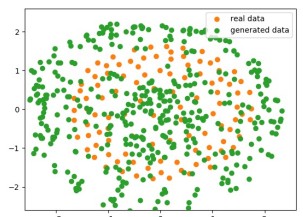

Figure 2: DSGAN generated boundary points of a swissroll.

Figure 3: Left: A circle which the density is low in the center. Right: Green points are generated by DSGAN.

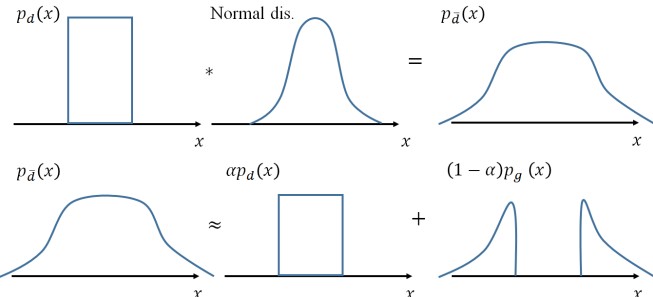

Figure 4: The illustration about generating boundary data around training data. First, the convolution of $p_d$ and normal distribution makes the density on boundary data be no longer zero. Second, we seek $p_g$ such that Eq. (1) holds, where the support set of $p_g$ is approximated by the difference of those between $p_{\bar{d}}$ and of $p_d$.

**Low-density samples generation** Fig. 2 illustrates that DSGAN is able to generate boundary samples on the 2D swissroll. Given the density function of the swissroll as $p_d$, we assign $p_{\bar{d}}$ as the convolution of $p_d$ and the normal distribution. Then, by applying DSGAN, we achieve our goal to generate boundary samples. The intuition of our idea is also illustrated by a 1D example in Fig. 4. In general, our idea will lead DSGAN to generate low-density samples like another example in Fig. 3. In this case, the density is low in the center of the circle, and our generator can not only create the boundary samples but also the samples located in low-density area.

**Difference-set generation** We also validate DSGAN on high dimensional dataset such as MNIST. In this example, we define $p_d$ be the distribution of digit "1" and $p_{\bar{d}}$ be the distribution contains both digits "1" and "7". Since the density $p_d(x)$ is high when $x$ is digit "1", the generator is prone to output digit "7" with high probability.

From the above results, we can observe two properties of generator distribution $p_g$: i) the higher density of $p_d(x)$, the lower density of $p_g(x)$; ii) $p_g$ prefers to output samples from high-density areas of $p_{\bar{d}}(x)$-$p_d(x)$. In those case studies, $\alpha = 0.8$ in Eq. (1) is used.

In the next section, we will show that the objective function is equivalent to minimizing the Jensen-Shannon divergence between the mixture distribution ($p_d$ and $p_g$) and $p_{\bar{d}}$ as $G$ and $D$ are given enough capacity. Furthermore, we provide a trick (see Appendix C in details) by reformulating the objective function (2) such that it is more stable to train DSGAN.

## 3 THEORETICAL RESULTS

There are two assumptions for subsequent proofs. First, in a nonparametric setting, we assume both generator and discriminator have infinite capacity. Second, $p_g$ is defined as the distribution of the samples drawn from $G(z)$ under $z \sim p_z$. We will first show the optimal discriminator given $G$ and then show that minimizing $V(G, D)$ via $G$ given the optimal discriminator is equivalent to minimizing the Jensen-Shannon divergence between $(1 - \alpha)p_g + \alpha p_d$ and $p_{\bar{d}}$.

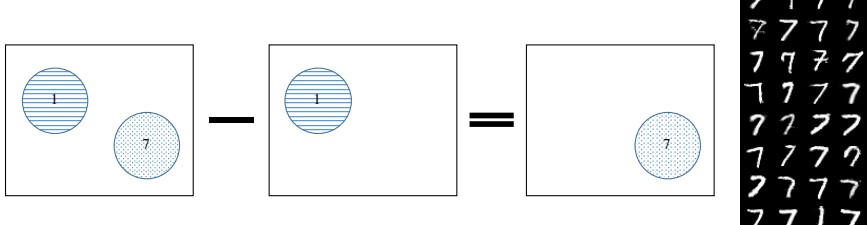

Figure 5: Illustration of difference-set seeking in MNIST.

Figure 6: DSGAN learn the difference between two sets.

**Proposition 1.** *For $G$ being fixed, the optimal discriminator $D$ is*

$$D_G^*(x) = \frac{p_d(x)}{p_d(x) + (1-\alpha)p_g(x) + \alpha p_d(x)}.$$

*Proof.* See Appendix B.1 in details. □

Moreover, $D$ can be considered to discriminate between samples from $p_{\bar{d}}$ and those from $((1-\alpha)p_g(x) + \alpha p_d(x))$. By replacing the optimal discriminator into $V(G, D)$, we obtain

$$
\begin{aligned}
C(G) &= \max_D V(G, D) \\
&= \mathbb{E}_{x \sim p_{\bar{d}}(x)}\left[\log D_G^*(x)\right] + (1-\alpha)\mathbb{E}_{z \sim p_z(z)}\left[\log\left(1 - D_G^*\left(G\left(z\right)\right)\right)\right] \\
&\qquad + \alpha \mathbb{E}_{x \sim p_d(x)}\left[\log\left(1 - D_G^*(x)\right)\right] \\
&= \mathbb{E}_{x \sim p_{\bar{d}}(x)}\left[\log D_G^*(x)\right] + \mathbb{E}_{x \sim (1-\alpha)p_g(x)+\alpha \sim p_d(x)}\left[\log\left(1 - D_G^*\left(x\right)\right)\right] \\
&= \mathbb{E}_{x \sim p_{\bar{d}}(x)}\left[\log \frac{p_{\bar{d}}(x)}{p_{\bar{d}}(x) + (1-\alpha)p_g(x) + \alpha p_d(x)}\right] \\
&\qquad + \mathbb{E}_{x \sim (1-\alpha)p_g(x)+\alpha \sim p_d(x)}\left[\log \frac{(1-\alpha)p_g(x) + \alpha p_d(x)}{p_{\bar{d}}(x) + (1-\alpha)p_g(x) + \alpha p_d(x)}\right],
\end{aligned}
\tag{3}
$$

where the third equality holds because of the linearity of expectation.

Actually, the previous results show the optimal solution of $D$ given $G$ being fixed in (3). Now, the next step is to find the optimal $G$ with $D_G^*$ being fixed.

**Theorem 1.** *Suppose $\alpha p_d(x) \leq p_{\bar{d}}(x)$ for all $x$'s. The global minimum of the virtual training criterion $C(G)$ is achieved if and only if $(1-\alpha)p_g(x) + \alpha p_d(x) = p_{\bar{d}(x)}$ for all $x$'s. At that point, $C(G)$ achieves the value $-\log 4$.*

*Proof.* See Appendix B.2 in details. □

The assumption, $\alpha p_d(x) \leq p_{\bar{d}}(x)$ for all $x$'s, in Theorem 1 may be impractical in real applications. We discuss that DSGAN still works well even though the assumption does not hold. There are two facts: i) given $D$, $V(G, D)$ is a convex function in $p_g$ and ii) Due to $\int_x p_g(x)dx = 1$, the set collecting all feasible solutions of $p_g$ is a convex set. In other words, there always exists a global minimum of $V(G, D)$ given $D$, but it may not be $-\log(4)$. In this following, we show that the support set of $p_g$ is contained within the difference of support sets of $p_{\bar{d}}$ and $p_d$ while achieving the global minimum such that we can generate the desired $p_g$ by designing appropriate $p_{\bar{d}}$.

**Proposition 2.** *Suppose $\alpha p_d(x) \geq p_{\bar{d}}$ for $x \in \mathrm{Supp}(p_d)$ and all density functions $p_d(x)$, $p_{\bar{d}}(x)$ and $p_g(x)$ are continuous. If the global minimum of the virtual training criterion $C(G)$ is achieved, then*

$$\mathrm{Supp}\left(p_g\right) \subseteq \mathrm{Supp}\left(p_{\bar{d}}\right) - \mathrm{Supp}(p_d).$$

*Proof.* See Appendix B.3 in details. □

In sum, the generator is prone to output samples located in high-density areas of $p_{\bar{d}}$ and low-density areas of $p_d$.

Another concern is the convergence of Algorithm 1.

**Proposition 3.** *The discriminator reaches its optimal value given $G$ in Algorithm 1, and $p_g$ is updated by minimizing*

$$\mathbb{E}_{x \sim p_{\bar{d}}(x)}\left[\log D_G^*(x)\right] + \mathbb{E}_{x \sim (1-\alpha)p_g(x) + \alpha \sim p_d(x)}\left[\log\left(1 - D_G^*(x)\right)\right].$$

*If $G$ and $D$ have enough capacity, then $p_g$ converges to* $\underset{p_g}{\operatorname{argmin}} \operatorname{JSD}\left(p_{\bar{d}} \parallel (1-\alpha)p_g + \alpha p_d\right)$.

*Proof.* See Appendix B.4 in details. □

## 4 APPLICATIONS

DSGAN can be applied to two applications: semi-supervised learning and adversarial training. In semi-supervised learning, DSGAN acts as a "bad generator", which creates complement samples in the feature space of real data. As for adversarial training, DSGAN generates adversarial examples located in the low-density areas of training data.

### 4.1 SEMI-SUPERVISED LEARNING

Semi-supervised learning (SSL) is a kind of learning model with the use of a small number of labeled data and a large amount of unlabeled data. The existing SSL works based on generative model (*e.g.,* VAE (Kingma et al. (2014)) and GAN (Salimans et al. (2016))) obtain good empirical results. Dai et al. (2017) theoretically show that good semi-supervised learning requires a bad GAN with the objective function:

$$\max_D \mathbb{E}_{x,y \sim \mathcal{L}} \log P_D\left(y \mid x, y \leq K\right) + \mathbb{E}_{x \sim p_d(x)} \log P_D\left(y \leq K \mid x\right) \\ + \mathbb{E}_{x \sim p_g(x)} \log P_g\left(K + 1 \mid x\right), \tag{4}$$

where $(x, y)$ denotes a pair of data and its corresponding label, $\{1, 2, \ldots, K\}$ denotes the label space for classification, and $\mathcal{L} = \{(x, y)\}$ is the label dataset. Moreover, in the semi-supervised settings, the $p_d$ in (4) is the distribution of the unlabeled data. Note that the discriminator $D$ in GAN also plays the role of classifier. If the generator distribution exactly matches the real data distribution (*i.e.*, $p_g = p_d$), then the classifier trained by the objective function (4) with the unlabeled data cannot have better performance than that trained by supervised learning with the objective function:

$$\max_D \mathbb{E}_{x,y \sim \mathcal{L}} \log P_D\left(y \mid x, y \leq K\right). \tag{5}$$

On the contrary, the generator is preferred to generate complement samples, which lie on low-density area of $p_d$. Under some mild assumptions, those complement samples help $D$ to learn correct decision boundaries in low-density area because the probabilities of the true classes are forced to be low on out-of-distribution areas.

The complement samples in Dai et al. (2017) are complicate to produce. We will demonstrate that DSGAN is easy to generate complement samples in Sec. 5.

### 4.2 ADVERSARIAL TRAINING

Deep neural networks have impacted on our daily life. Neural networks, however, are vulnerable to adversarial examples, as evidenced in recent studies (Papernot et al. (2016))(Carlini & Wagner (2017)). Thus, there has been significant interest in how to enhance the robustness of neural networks. Unfortunately, if the adversary has full access to the network, namely white-box attack, a complete defense strategy has not yet been found.

Athalye et al. (2018) surveyed the state-of-the-art defense strategies and showed that adversarial training (Madry et al. (2018)) is more robust than other strategies. Given a trained classifier $C$ parameterized by $\theta$ and a loss function $\ell(x; y; C_\theta)$, adversarial training solves a min-max game,

where the first step is to find adversarial examples within $\epsilon$-ball for maximizing the loss, and the second step is to train the model for minimizing the loss, given adversarial examples. Specifically, the objective (Madry et al. (2018)) is

$$\underset{\theta}{\operatorname{argmin}}\, \mathbb{E}_{(x,y)\sim\mathcal{L}} \left[ \max_{\delta\in[-\epsilon,\,\epsilon]^N} \ell\left(x; y; C_\theta\right) \right]. \tag{6}$$

The authors used projected gradient descent (PGD) to find adversarial examples by maximizing the inner optimization.

Instead of relying on PGD, our DSGAN generates adversarial examples directly, which are combined into real training data to fine-tune $C_\theta$. $\epsilon$-ball in terms of $\ell_2$ or $\ell_{\inf}$ can be intuitively incorporated into the generation of adversarial examples.

## 5 EXPERIMENTS

In this section, we demonstrate the empirical results about semi-supervised learning and adversarial training in Sec. 5.1 and Sec. 5.2, respectively.

Note that, the training procedure of DSGAN can be improved by other extensions of GANs such as WGAN (Arjovsky et al. (2017)), WGAN-GP (Gulrajani et al. (2017)), EBGAN (Zhao et al. (2017)), LSGAN (Mao et al. (2017)) and etc. We use the idea of WGAN-GP in our method such that DSGAN is stable in training and suffers less mode collapse.

### 5.1 DSGAN IN SEMI-SUPERVISED LEARNING

Following the previous works, we apply the proposed DSGAN in semi-supervised learning on three benchmark datasets, including MNIST (LeCun et al. (1998)), SVHN (Netzer et al. (2011)), and CIFAR-10 (Krizhevsky (2009)).

We first introduce how DSGAN generates complement samples in the feature space. Specifically, Dai et al. (2017) proved that if complement samples generated by $G$ can satisfy the following two assumptions in (7) and (8):

$$\forall x \sim p_g(x), 0 > \max_{1\leq i\leq K} w_i^T f(x), \text{ and } \forall x \sim p_d(x), 0 < \max_{1\leq i\leq K} w_i^T f(x) \tag{7}$$

where $f$ is the feature extractor and $w_i$ is the linear classifier for the $i^{th}$ class and

$$\forall x_1 \sim \mathcal{L}, x_2 \sim p_d(x), \exists x_g \sim p_g(x) \text{ s.t. } f(x_g) = \beta f(x_1) + (1-\beta)f(x_2) \text{ with } \beta \in [0,\,1], \tag{8}$$

then all unlabeled data will be classified correctly via the objective function (4).

The assumption in (8) implies the complement samples have to be at the space created by linear combination of labeled and unlabeled data. Besides, they cannot fall into the real data distribution $p_d$ due to the assumption (7). In order to have DSGAN generate such samples, we let the samples of $p_{\bar{d}}$ be the linear combination of those from $\mathcal{L}$ and $p_d$. Since $p_g(x) \approx \dfrac{p_{\bar{d}}(x) - \alpha p_d(x)}{1-\alpha}$, the $p_g$ will tend to match $p_{\bar{d}}$ while the term $-\alpha p_d$ ensures that samples from $p_g$ do not belong to $p_d$. Thus, $p_g$ satisfies both assumptions in (7) and (8).

In practice, we parameterized $f$ and all the $w$ together as a neural network. The details of the experiments, including the network models, can be found in Appendix D.

### 5.1.1 MNIST, SVHN, AND CIFAR-10

For evaluating the semi-supervised learning task, we used 60000/ 73257/ 50000 samples and 10000/ 26032/ 10000 samples from the MNIST/ SVHN/ CIFAR-10 dataset for training and testing, respectively. Due to the semi-supervised setting, we randomly chose 100/ 1000/ 4000 samples from the training samples as the MNIST/ SVHN/ CIFAR-10 labeled dataset, and the amount of labeled data for all classes are equal.

Our criterion to determine the hyperparameters are in Appendix D.1. We perform testing with 10/ 5/ 5 runs on MNIST/ SVHN/ CIFAR-10 based on the selected hyperparameters and randomly selected labeled dataset. Following Dai et al. (2017), the results are recorded as the mean and standard deviation of number of errors from each run.

### 5.1.2 Main results

First, the hyperparameters we chose is depicted in Table 3 in Appendix D.1. Second, the results obtained from our DSGAN and the state-of-the-art methods on three benchmark datasets are depicted in Table 1.

It can be observed that our results can compete with state-of-the-art methods on the three datasets. Moreover, in comparison with Dai et al. (2017), our methods don't need to rely on an additional density estimation network PixelCNN++ (Salimans et al. (2017)). Although PixelCNN++ is one of the best density estimation network, it cannot estimate the density in the feature space, which is dynamic during training. This drawback make the models in Dai et al. (2017) cannot completely fulfill the assumptions in their paper.

Table 1: Comparison of semi-supervised learning between our DSGAN and other state-of-the-art results. For fair comparison, we only consider the GAN-based methods. $*$ indicates the use of the same architecture of classifier. $\dagger$ indicates a larger architecture of classifier. $\ddagger$ indicates the use of data augmentation.

| Methods | MNIST (# errors) | SVHN (% of errors) | CIFAR-10 (% of errors) |
|---|---|---|---|
| CatGAN (Springenberg (2016)) | $191(\pm10)$ | - | $19.58(\pm0.46)$ |
| TripleGAN$^\dagger$ (Li et al. (2017)) | $91(\pm58)$ | $5.77(\pm0.17)$ | $16.99(\pm0.36)$ |
| FM$^*$ (Salimans et al. (2016)) | $93(\pm6.5)$ | $8.11(\pm1.3)$ | $18.63(\pm1.32)$ |
| badGAN$^*$ (Dai et al. (2017)) | $79.5(\pm9.8)$ | $4.25(\pm0.03)$ | $14.41(\pm0.30)$ |
| CT-GAN$^\ddagger$ (Wei et al. (2018)) | - | - | $9.98(\pm0.21)$ |
| Our method$^*$ | $82.7(\pm4.6)$ | $5.01(\pm0.14)$ | $15.08(\pm0.24)$ |

Our results in Table 1 is slightly inferior to the best record of badGAN (Dai et al. (2017)) but outperforms other approaches. In comparison with them, there is a probable reason to explain the slightly inferior performance in the following. Since the patterns of images are complicated, the generator without enough capacity is not able to learn our desired distribution, which is the distribution meets the conditions in (7) and (8). However, this problem will be attenuated with the improvements of GAN, and our models benefit from them. In badGAN, they rely on the feature matching in their objective function. No matter how they change the divergence criterion on two distributions, feature matching still let them learn the distribution matching the first-order statistics, so they cannot totally get the advantage of the progress of GANs.

## 5.2 DSGAN in Adversarial Training

Our proposed DSGAN is capable to be used to improve the robustness of the classifier against adversarial examples. In the experiments, we mainly validate DSGAN on CIFAR-10, which is widely used in adversarial training.

Recall that the objective function (6) requires finding adversarial examples to maximize the classification error $\ell(\cdot)$. Adversarial examples usually locate on the low-density area of $p_d$ and are generated from labeled data via gradient descent. Instead of using gradient descent, we aim to generate adversarial examples via GAN. By assigning $p_{\bar{d}}$ as the convolution of $p_d$ and uniform distribution, samples from $p_g$ will locate on the low-density area of $p_d$. Furthermore, the distortion $\epsilon$ is directly related to the range of uniform distribution. It, however, may be impractical for training the generator for each class. Thus, we propose a novel semi-supervised adversarial learning approach here.

Three stages are required to train our model: First, we train a baseline classifier on all the training data. All the training data are labeled and represent samples from $\mathcal{L}$ in (9). Second, we train a generator to generate adversarial examples and treat these adversarial examples as additional unlabeled training data ($x \sim p_g$ in (9)). Third, we fine-tune the classifier $C_\theta$ with all training data and the data produced by the generator via minimizing the following objective:

$$\underset{\theta}{\arg\min}\, \mathbb{E}_{(x,y)\sim\mathcal{L}}\left[\ell\left(x;y;C_\theta\right)\right] + w \cdot \mathbb{E}_{x_g\sim p_g(x)}\left[\mathcal{H}\left(C_\theta(x_g)\right)\right], \quad (9)$$

where the first term is a typical supervised loss such as cross-entropy loss and the second term is the entropy loss $\mathcal{H}$ of generated unlabeled samples corresponding to the classifier, meaning that

we would like the classifier to confidently classify the generated samples. In other words, if an adversarial example $x_g$ is the closest to one of labeled data $x$, it should be classified into the class of $x$. Thus, the additional entropy loss will prevent our model from the attack by adversarial examples.

Furthermore, in (9), one can view the weight $w$ is the trade-off between the importance of labeled data in high-density area and unlabeled data in low-density area. If $w$ is 0, the model might be prone to classify correctly only on the labeled data. When increasing $w$, the model will place more emphasis on unlabeled data. Since the unlabeled data acts as adversarial examples, therefore, the classifier is more robustness.

### 5.2.1 Experiments settings

We evaluate the trained models against a range of adversaries, where the distortion is evaluated by $\ell_2$-norm or $\ell_{\text{inf}}$-norm. The adversaries include:

- White-box attacks with Fast Gradient Sign Method (FGSM) (Goodfellow et al. (2015)) using $\ell_{\text{inf}}$-norm.
- White-box attacks with PGD (Kurakin et al. (2017)) using $\ell_{\text{inf}}$-norm.
- White-box attacks with Deepfool (Moosavi-Dezfooli et al. (2016)) using $\ell_2$-norm and $\ell_{\text{inf}}$-norm.

According to different adversaries, we generate 10000 adversarial examples from testing data and calculate the accuracy of the model after attacking. The accuracy is record as the probability that adversarial examples fail to attack when the distortion created by attacking algorithm cannot exceed a maximum value. We also train our models with different ranges of uniform distribution. The experimental detail can be found in Appendix E.

To validate our method, we propose two kinds of baseline networks. One is a baseline classifier we train in the first stage, which is a typical classifier trained by all data. The other one is the model with noisy inputs. Adding noise to the input is a prevalent strategy to train a classifier and it is also able to protect the neighborhood of the training data. For fare comparison to our method, uniform noise is used in the second baseline model.

### 5.2.2 Main results

Fig. 7 demonstrates that our models exhibit stronger robustness among all the adversaries. $w$ is set to 10 in this figure, other results with different $w$ are displayed in the Appendix E and we claim that our method can outperform other baselines in a wide range of values of $w$. We notice that the model benefits from controlling the weight $w$. When we increase the $w$ from 1 to 3, and then from 3 to 10, the robustness keeps becoming stronger.

Our second baseline models have the similar intuition with our method, they propagate the label information to the neighborhood of each data point by introducing the noise to inputs. This strategy can improve the accuracy and the robustness of the model. Nevertheless, the training data distribution after applying noise can be viewed as a smoother version of original distribution. Most of samples still are locate on high-density area of original distribution. Due to this reason, the second baseline models cannot emphasize low-density samples via $w$ like the proposed model, leading to the inferior robustness.

Our method relies on a generator to produce low-density data. The generated samples help our model to put decision boundary outside low-density area. Thus, the model can resist adversarial attacks with larger distortion theoretically. It's worth mentioning that our method is able to combine with the idea of second baseline to the supervised term in (9) and the performance might be improved.

## 6 Related Works

We introduce related works about generating unseen data.

Yu et al. (2017) proposed a method to generate samples of unseen classes in the unsupervised manner via an adversarial learning strategy. But, it requires solving an optimization problem for each sample, which may lead to high computation cost. On the contrary, DSGAN has the capability to create infinite diverse unseen samples. Hou et al. (2018) presented a new GAN architecture that can learn both distributions of unseen data from part of seen data and unlabeled data. But, unlabeled

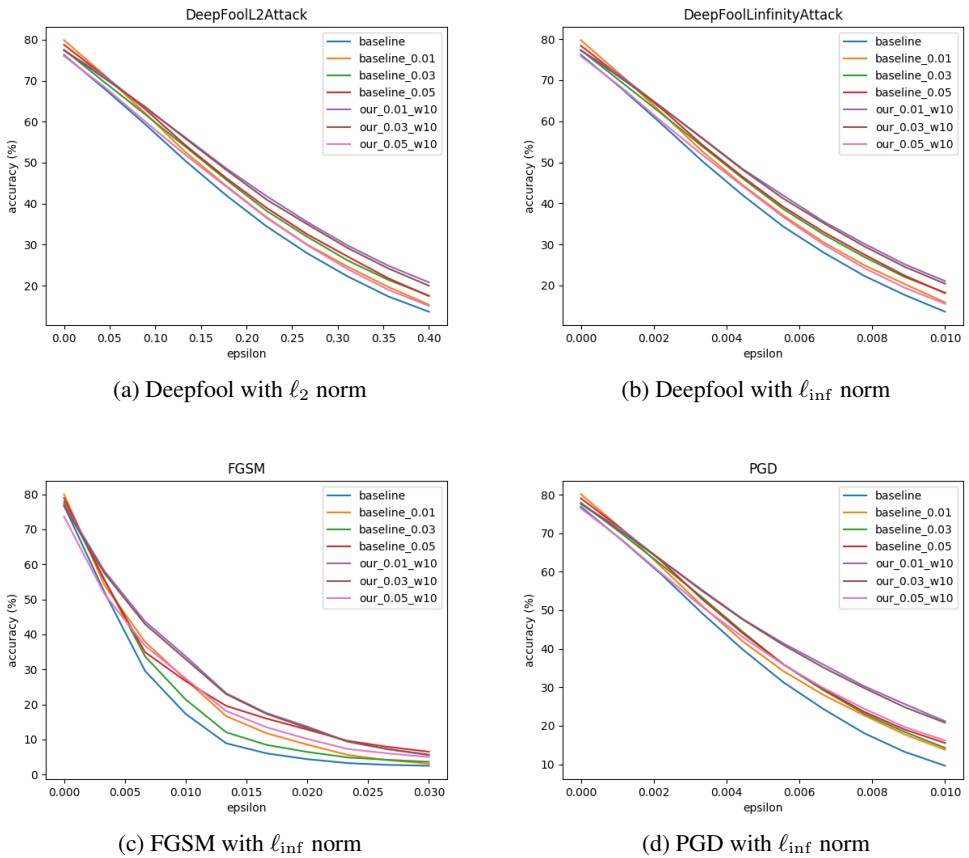

Figure 7: The accuracy of baseline and our models after attacks. Blue line is first baseline model. Orange, green and red lines are second baseline models with different range of uniform noise. Purple, brown and pink lines are our methods. The float number also indicates the range of noise where "w10" means that $w$ in (9) is set to 10. The epsilon means that the $\ell_2$ (or $\ell_{\text{inf}}$) norm between original image (pixel values are normalized to range $[-0.5, 0.5]$) and corresponding adversarial example.

data must be a mixture of seen and unseen samples; DSGAN does not require any unseen data instead. Both Dai et al. (2017) aim to generate complementary samples (or out-of-distribution sample) but assume that in-distribution can be estimated by a pretrained model such as PixelCNN++, which might be difficult and expensive to train. Lee et al. (2018) use a simple classifier to replace the role of PixelCNN++ in Dai et al. (2017) such that the training is much easier and more suitable. Nevertheless, their method only focuses on generates unseen data surrounding low-density area of seen data, but DSGAN is more flexible to generate different kinds of unseen data (*e.g.,* the linear combination of seen data shown in Sec.5.1). Besides, their method needs the label information of data while ours is fully unsupervised.

## 7 CONCLUSIONS

In this paper, we propose DSGAN that can produce samples from the target distribution based on the assumption that the density of target distribution can be the difference between the densities of any two distributions. DSGAN is useful in the environment when the samples from the target distribution are more difficult to collect than those from the two known distributions. We demonstrate that DSGAN is really applicable to, for example, semi-supervised learning and adversarial training. Empirical and theoretical results are provided to validate the effectiveness of DSGAN. Finally, because DSGAN is developed based on traditional GAN, it is easy to extend any improvements of traditional GAN to DSGAN.

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

## APPENDIX

## A    APPLICATIONS ABOUT UNSEEN DATA

We show key applications regarding as unseen data as follow:

## B    PROOFS

### B.1    PROOF OF THE PROPOSITION 1

*Proof.* Given any generator $G$, the training criterion for the discriminator $D$ is to maximize the quantity $V(G, D)$:

$$
\begin{aligned}
V(G, D) &= \int_x p_{\bar{d}}(x) \log\left(D\left(x\right)\right) dx + (1 - \alpha) \int_z p_z(z) \log\left(1 - D\left(G\left(z\right)\right)\right) dz \\
&\quad + \alpha \int_x p_d(x) \log\left(1 - D\left(x\right)\right) dx \\
&= \int_x p_{\bar{d}}(x) \log\left(D\left(x\right)\right) dx + (1 - \alpha) \int_x p_g(x) \log\left(1 - D\left(x\right)\right) dz \\
&\quad + \alpha \int_x p_d(x) \log\left(1 - D\left(x\right)\right) dx \\
&= \int_x p_{\bar{d}}(x) \log\left(D\left(x\right)\right) + ((1 - \alpha)p_g(x) + \alpha p_d(x)) \log\left(1 - D\left(x\right)\right) dx.
\end{aligned}
$$

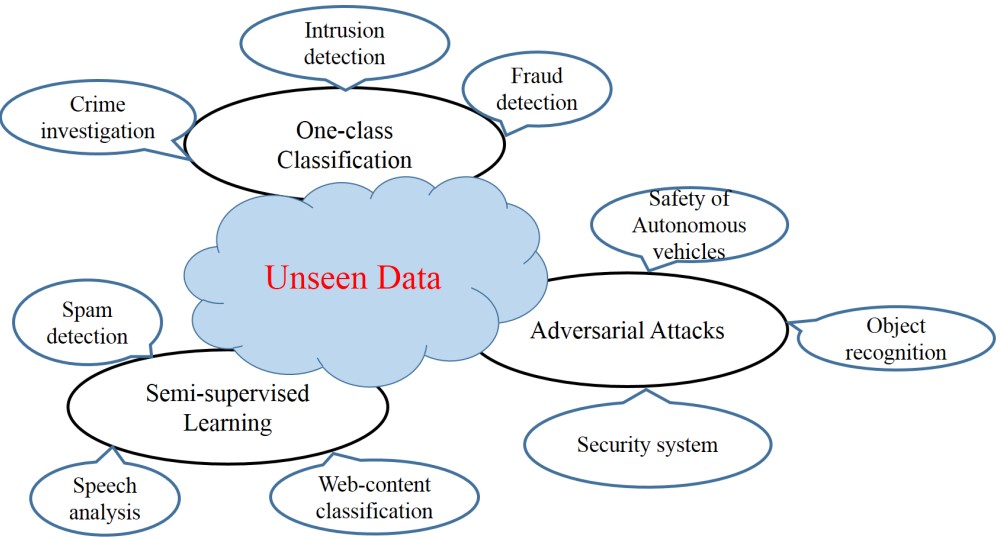

Figure 8: The illustration of applications related to unseen data.

For any $(a, b) \in \mathbb{R}^2 \backslash \{0, \, 0\}$, the function $a \log(y) + b \log(1 - y)$ achieves its maximum in $[0, \, 1]$ at $y = \frac{a}{a+b}$. The discriminator only needs to be defined within $\text{Supp}(p_{\bar{d}}) \bigcup \text{Supp}(p_d) \bigcup \text{Supp}(p_g)$. We complete this proof. $\qquad\square$

### B.2 PROOF OF THEOREM 1

*Proof.* We start from

$$
\begin{aligned}
(3) = {} & -\log(4) + \mathbb{E}_{x \sim p_{\bar{d}}(x)} \left[ \log \frac{2 p_{\bar{d}}(x)}{p_{\bar{d}}(x) + (1 - \alpha) p_g(x) + \alpha p_d(x)} \right] \\
& + \mathbb{E}_{x \sim (1-\alpha) p_g(x) + \alpha \sim p_d(x)} \left[ \log \frac{2 \left( (1 - \alpha) p_g(x) + \alpha p_d(x) \right)}{p_{\bar{d}}(x) + (1 - \alpha) p_g(x) + \alpha p_d(x)} \right] \\
= {} & -\log(4) + \text{KL} \left( p_{\bar{d}} \,\Big\|\, \frac{p_{\bar{d}} + (1 - \alpha) p_g + \alpha p_d}{2} \right) \\
& + \text{KL} \left( (1 - \alpha) p_g(x) + \alpha p_d \,\Big\|\, \frac{p_{\bar{d}} + (1 - \alpha) p_g + \alpha p_d}{2} \right) \\
= {} & -\log(4) + 2 \, \text{JSD} \left( p_{\bar{d}} \,\|\, (1 - \alpha) p_g + \alpha p_d \right),
\end{aligned}
$$

where KL is the Kullback-Leibler divergence and JSD is the Jensen-Shannon divergence. The JSD returns the minimal value, which is 0, iff both distributions are the same, namely $p_{\bar{d}} = (1 - \alpha) p_g + \alpha p_d$. Because $p_g(x)$'s are always non-negative, it should be noted both distributions are the same only if $\alpha p_d(x) \le p_{\bar{d}}(x)$ for all $x$'s. We complete this proof. $\qquad\square$

### B.3 Proof of Proposition 2

*Proof.* Recall

$$C(G) = \int_x p_{\bar{d}}(x) \log\left(\frac{p_{\bar{d}}(x)}{p_{\bar{d}}(x) + (1-\alpha)p_g(x) + \alpha p_d(x)}\right)$$
$$+ ((1-\alpha)p_g(x) + \alpha p_d(x)) \log\left(\frac{(1-\alpha)p_g(x) + \alpha p_d(x)}{p_{\bar{d}}(x) + (1-\alpha)p_g(x) + \alpha p_d(x)}\right) dx$$

$$= \int_x S(p_g; x) dx$$

$$= \int_{x \in \mathrm{Supp}(p_{\bar{d}}) - \mathrm{Supp}(p_d)} S(p_g; x) dx + \int_{x \in \mathrm{Supp}(p_d)} S(p_g; x) dx$$

$S(p_g; x)$ is to simplify the notations inside the integral. For any $x$, $S(p_g; x)$ in $p_g(x)$ is non-increasing and $S(p_g; x) \leq 0$ always holds. Specifically, $S(p_g; x)$ is decreasing along the increase of $p_g(x)$ if $p_{\bar{d}}(x) > 0$; $S(p_g; x)$ attains the maximum value, zero, for any $p_g(x)$ if $p_{\bar{d}}(x) = 0$. Since DSGAN aims to minimize $C(G)$ with the constraint $\int_x p_g(d) dx = 1$, the solution attaining the global minima must satisfy $p_g(x) = 0$ if $p_{\bar{d}}(x) = 0$; otherwise, there exists another solution with smaller value of $C(G)$. Thus, $\mathrm{Supp}(p_g) \subseteq \mathrm{Supp}(p_{\bar{d}})$.

Furthermore, $T(p_g; x) = \frac{\partial S(p_g; x)}{\partial p_g(x)} = \log\left(\frac{(1-\alpha)p_g(x) + \alpha p_d(x)}{p_{\bar{d}}(x) + (1-\alpha)p_g(x) + \alpha p_d(x)}\right)$ is increasing on $p_g(x)$ and converges to 0. When $x \in \mathrm{Supp}(p_{\bar{d}}) \bigcap \mathrm{Supp}(p_d)$, $T(p_g; x) \geq \log \frac{1}{2}$ always holds due to the assumption $\alpha p_d(x) \geq p_{\bar{d}}(x)$. Based on the following expression,

$$\int_{x \in \mathrm{Supp}(p_{\bar{d}}) - \mathrm{Supp}(p_d)} p_{\bar{d}}(x) dx + \int_{x \in \mathrm{Supp}(p_d)} p_{\bar{d}}(x) dx = 1$$

$$\Rightarrow \int_{x \in \mathrm{Supp}(p_{\bar{d}}) - \mathrm{Supp}(p_d)} p_{\bar{d}}(x) dx \geq 1 - \int_{x \in \mathrm{Supp}(p_d)} \alpha p_d(x) dx$$

$$\Rightarrow \int_{x \in \mathrm{Supp}(p_{\bar{d}}) - \mathrm{Supp}(p_d)} p_{\bar{d}}(x) dx \geq 1 - \alpha$$

$$\Rightarrow \int_{x \in \mathrm{Supp}(p_{\bar{d}}) - \mathrm{Supp}(p_d)} p_{\bar{d}}(x) dx \geq \int_{x \in \mathrm{Supp}(p_{\bar{d}}) - \mathrm{Supp}(p_d)} (1-\alpha)p_g(x) dx,$$

the last inequality implies that there exists a solution such that $(1-\alpha)p_g(x) \leq p_{\bar{d}(x)}$ for $x \in \mathrm{Supp}(p_{\bar{d}}) - \mathrm{Supp}(p_d)$ with $\int_{x \in \mathrm{Supp}(p_{\bar{d}}) - \mathrm{Supp}(p_d)} p_g(d) dx = 1$ In this case, $T(p_g; x) < \log \frac{1}{2}$ for $x \in \mathrm{Supp}(p_{\bar{d}}) - \mathrm{Supp}(p_d)$. We complete this proof. $\square$

### B.4 Proof of Proposition 3

*Proof.* Consider $V(G, D) = U(p_g, D)$ as a function of $p_g$. By the proof idea of Proposition 2 in Goodfellow et al. (2014), if $f(x) = \sup_{\alpha \in \mathcal{A}} f_\alpha(x)$ and $f_\alpha(x)$ is convex in $x$ for every $\alpha$, then $\partial f_\beta(x) \in \partial f$ if $\beta = \mathrm{argsup}_{\alpha \in \mathcal{A}} f_\alpha(x)$. In other words, if $\sup_D V(G, D)$ is convex in $p_g$, the subderivatives of $\sup_D V(G, D)$ includes the derivative of the function at the point, where the maximum is attained, implying the convergence with sufficiently small updates of $p_g$. We complete this proof. $\square$

## C Tricks for Stable Training

We provide a trick to stabilize the training procedure by reformulating the objective function. Specifically, $V(G, D)$ in (2) is reformulated as:

$$V(G, D) = \int_x p_{\bar{d}}(x) \log(D(x)) + ((1-\alpha)p_g(x) + \alpha p_d(x)) \log(1 - D(x)) dx$$
$$= \mathbb{E}_{x \sim p_{\bar{d}}(x)}[\log D(x)] + \mathbb{E}_{x \sim (1-\alpha)p_g(x) + \alpha \sim p_d(x)}[\log(1 - D(x))]. \tag{10}$$

Instead of sampling a mini-batch of $m$ samples from $p_z$ and $p_d$ in Algorithm 1, $(1 - \alpha)m$ and $\alpha m$ samples from both distributions are required, respectively. The computation cost in training can be reduced due to fewer samples. Furthermore, although (10) is equivalent to (2) in theory, we find that the training using (10) achieves better performance than using (2) via empirical validation in Table 2. We conjecture that the equivalence between (10) and (2) is based on the linearity of expectation, but mini-batch stochastic gradient descent in practical training may lead to the different outcomes.

Table 2: Comparing the semi-supervised learning results on MNIST whether to use the sampling tricks.

| Methods | MNIST (# errors) |
|---|---|
| Our method w/o tricks | $91.0(\pm 7.0)$ |
| Our method w/ tricks | $82.7(\pm 4.6)$ |

## D    EXPERIMENTAL DETAILS FOR SEMI-SUPERVISED LEARNING

### D.1    HYPERPARAMETERS

The hyperparameters were chosen to make our generated samples consistent with the assumptions in (7) and (8). However, in practice, if we make all the samples produced by the generator following the assumption in (8), then the generated distribution is not close to the true distribution, even a large margin between them existing in most of the time, which is not what we desire. So, in our experiments, we make a concession that the percentage of generated samples, which accords with the assumption, is around $90\%$. To meet this objective, we tune the hyperparameters. Table 3 shows our setting of hyperparameters, where $\beta$ is defined in (8).

Table 3: Hyperparameters in semi-supervised learning.

| Hyperparameters | MNIST | SVHN | CIFAR-10 |
|---|---|---|---|
| $\alpha$ | 0.8 | 0.8 | 0.5 |
| $\beta$ | 0.3 | 0.3 | 0.1 |

### D.2    ARCHITECTURE

In order to fairly compare with other methods, our generators and classifiers for MNIST, SVHN, and CIFAR-10 are same as in Salimans et al. (2016) and Dai et al. (2017). However, different from previous works that have only a generator and a discriminator, we design an additional discriminator in the feature space, and it's architecture is similar across all datasets with only the difference in the input dimensions. Following Dai et al. (2017), we also define the feature space as the input space of the output layer of discriminators.

Compared to SVHN and CIFAR-10, MNIST is a simple dataset as it is only composed of fully connected layers. Batch normalization (BN) or weight normalization (WN) is used to every layer to stable training. Moreover, Gaussian noise is added before each layer in the classifier, as proposed in Rasmus et al. (2015). We find that the added Gaussian noise exhibits positive effect for semi-supervised learning and keep to use it. The architecture is shown in Table 4.

Table 5 and Table 6 are models for SVHN and CIFAR-10, respectively, and these models are almost the same except for some implicit differences, *e.g.*, the number of convolutional filters and types of dropout. In these tables, given a dropping rate, "Dropout" is a normal dropout in that the elements of input tensor are randomly set to zero while Dropout2d is a dropout only applied on the channels to randomly zero all the elements.

Furthermore, the training procedure alternates between $k$ steps of optimizing $D$ and one step of optimizing $G$. We find that $k$ in Algorithm 1 is a key role to the problem of mode collapse for different applications. For semi-supervised learning, we set $k = 1$ for all datasets.

Table 4: Network architectures for semi-supervised learning on MNIST. (GN: Gaussian noise)

| Generator $G$ | Discriminator $D$ | Classifier $C$ |
|---|---|---|
| Input $z \in \mathbb{R}^{100}$ from unif(0, 1) | Input $28 \times 28$ gray image | Input $28 \times 28$ gray image |
| $100 \times 500$ FC layer with BN Softplus $500 \times 500$ FC layer with BN Softplus $500 \times 784$ FC layer with WN Sigmoid | $250 \times 400$ FC layer ReLU $400 \times 200$ FC layer ReLU $200 \times 100$ FC layer ReLU $100 \times 1$ FC layer | GN, std = 0.3 $784 \times 1000$ FC layer with WN ,ReLU GN, std = 0.5 $1000 \times 500$ FC layer with WN, ReLU GN, std = 0.5 $500 \times 250$ FC layer with WN, ReLU GN, std = 0.5 $250 \times 250$ FC layer with WN, ReLU GN, std = 0.5 $250 \times 250$ FC layer with WN, ReLU $250 \times 10$ FC layer with WN |

Table 5: Architectures of generator and discriminator for semi-supervised learning on SVHN and CIFAR-10. $N$ was set to 128 and 192 for SVHN and CIFAR-10, respectively.

| Generator $G$ | Discriminator $D$ |
|---|---|
| Input $z \in \mathbb{R}^{100}$ from unif(0, 1) | Input $32 \times 32$ RGB image |
| $100 \times 8192$ FC layer with BN, ReLU Reshape to $4 \times 4 \times 512$ $5 \times 5$ conv. transpose 256 stride = 2 with BN, ReLU $5 \times 5$ conv. transpose 128 stride = 2 with BN, ReLU $5 \times 5$ conv. transpose 3 stride = 2 with WN, Tanh | $N \times 400$ FC layer, ReLU $400 \times 200$ FC layer, ReLU $200 \times 100$ FC layer, ReLU $100 \times 1$ FC layer |

Table 6: The architecture of classifiers for semi-supervised learning on SVHN and CIFAR-10. (GN: Gaussian noise, lReLU(leak rate): LeakyReLU(leak rate))

| Classifier $C$ for SVHN | Classifier $C$ for CIFAR-10 |
|---|---|
| Input $32 \times 32$ RGB image | Input $32 \times 32$ RGB image |
| GN, std = 0.05 Dropout2d, dropping rate = 0.15 $3 \times 3$ conv. 64 stride = 1 with WN, lReLU(0.2) $3 \times 3$ conv. 64 stride = 1 with WN, lReLU(0.2) $3 \times 3$ conv. 64 stride = 2 with WN, lReLU(0.2) Dropout2d, dropping rate = 0.5 $3 \times 3$ conv. 128 stride = 1 with WN, lReLU(0.2) $3 \times 3$ conv. 128 stride = 1 with WN, lReLU(0.2) $3 \times 3$ conv. 128 stride = 2 with WN, lReLU(0.2) Dropout2d, dropping rate = 0.5 $3 \times 3$ conv. 128 stride = 1 with WN, lReLU(0.2) $1 \times 1$ conv. 128 stride = 1 with WN, lReLU(0.2) $1 \times 1$ conv. 128 stride = 1 with WN, lReLU(0.2) Global average Pooling $128 \times 10$ FC layer with WN | GN, std = 0.05 Dropout2d, dropping rate = 0.2 $3 \times 3$ conv. 96 stride = 1 with WN, lReLU(0.2) $3 \times 3$ conv. 96 stride = 1 with WN, lReLU(0.2) $3 \times 3$ conv. 96 stride = 2 with WN, lReLU(0.2) Dropout, dropping rate = 0.5 $3 \times 3$ conv. 192 stride = 1 with WN, lReLU(0.2) $3 \times 3$ conv. 192 stride = 1 with WN, lReLU(0.2) $3 \times 3$ conv. 192 stride = 2 with WN, lReLU(0.2) Dropout, dropping rate = 0.5 $3 \times 3$ conv. 192 stride = 1 with WN, lReLU(0.2) $1 \times 1$ conv. 192 stride = 1 with WN, lReLU(0.2) $1 \times 1$ conv. 192 stride = 1 with WN, lReLU(0.2) Global average Pooling $192 \times 10$ FC layer with WN |

# E    EXPERIMENTAL DETAILS FOR ADVERSARIAL TRAINING

The size of labeled data for CIFAR-10 is 50000 and we balance the number of data for each class.

In our experiments, as for the second stage, we train DSGAN for $50$ epochs in Algorithm 1 to generate our adversarial examples. In the third stage, we finetune the baseline classifier for $50$ epochs.

In the experiments for adversarial training on CIFAR-10, the generator and discriminator are the same as those in semi-supervised learning. The architecture is described in Table 5 and the classifier is modified from the one shown in Table 6. First, we get rid of all the dropouts and Gaussian noise so that we can compare among different models with less randomness. Moreover, we decrease the number of layers in the original model, simply intending to accelerate training. The number of layers following the feature space is increased to $3$. Because we apply our method in the feature space, the sub-network after feature space should be non-linear so that it can correctly classify generated data. The architecture is described in Table 7. Furthermore, $k$ is assigned to $5$ in all experiments.

We show more results in Fig. 9 with $w = 1$ and 10 with $w = 3$.

Table 7: The architecture of classifier for adversarial training on CIFAR-10. (lReLU(leak rate): LeakyReLU(leak rate))

| Classifier $C$ for CIFAR-10 |
| --- |
| Input $32 \times 32$ RGB image |
| $3 \times 3$ conv. 96 stride = 1 with WN, lReLU(0.2) |
| $3 \times 3$ conv. 96 stride = 2 with WN, lReLU(0.2) |
| Dropout, dropping rate = 0.5 |
| $3 \times 3$ conv. 192 stride = 1 with WN, lReLU(0.2) |
| $3 \times 3$ conv. 192 stride = 2 with WN, lReLU(0.2) |
| Dropout, dropping rate = 0.5 |
| $3 \times 3$ conv. 192 stride = 1 with WN, lReLU(0.2) |
| $1 \times 1$ conv. 192 stride = 1 with WN, lReLU(0.2) |
| Global average Pooling |
| |
| $192 \times 192$ FC layer with WN, lReLU(0.2) |
| $192 \times 192$ FC layer with WN, lReLU(0.2) |
| $192 \times 192$ FC layer with WN, lReLU(0.2) |
| $192 \times 10$ FC layer with WN |

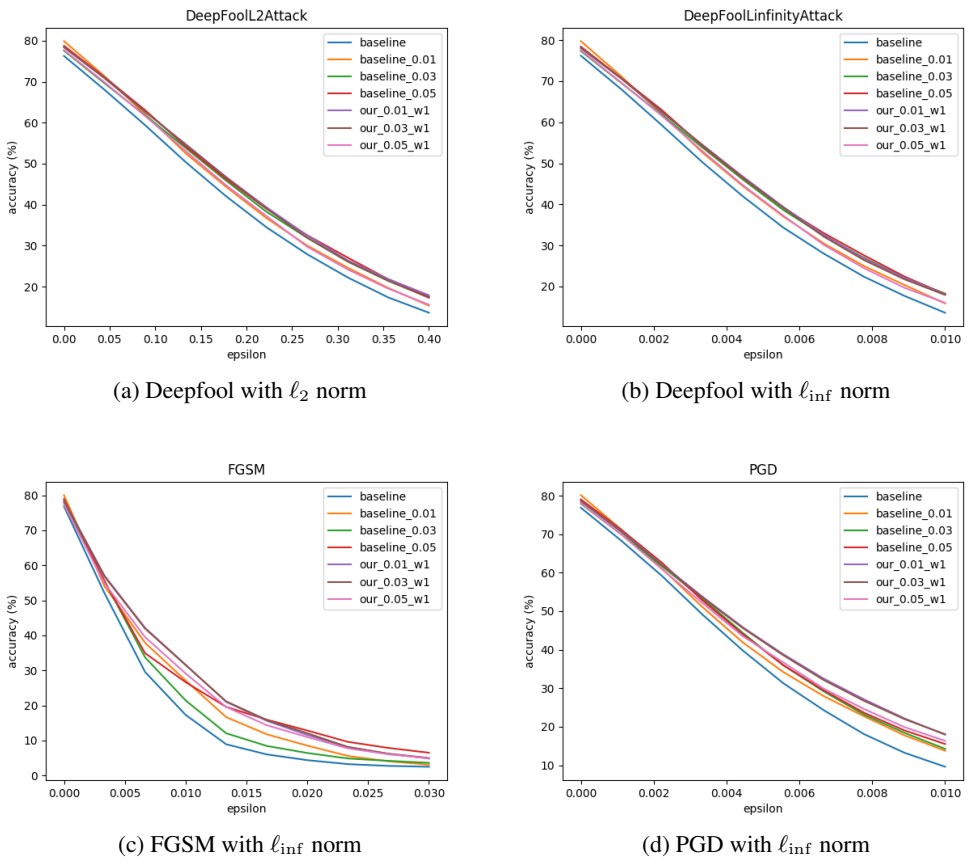

(a) Deepfool with $\ell_2$ norm

(b) Deepfool with $\ell_{\inf}$ norm

(c) FGSM with $\ell_{\inf}$ norm

(d) PGD with $\ell_{\inf}$ norm

Figure 9: The setting is the same with Fig. 7 unless $w = 1$.

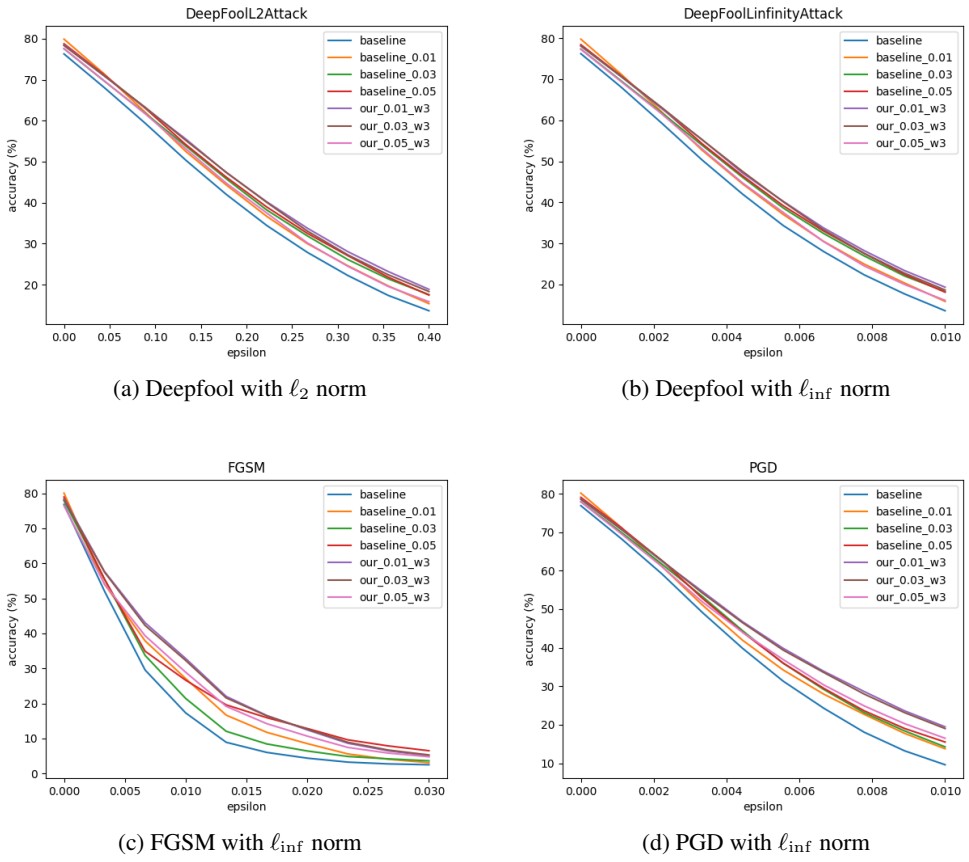

(a) Deepfool with $\ell_2$ norm

(b) Deepfool with $\ell_{\inf}$ norm

(c) FGSM with $\ell_{\inf}$ norm

(d) PGD with $\ell_{\inf}$ norm

Figure 10: The setting is the same with Fig. 7 unless $w = 3$.

