# OpenReview forum: "Difference-Seeking Generative Adversarial Network"
_ICLR.cc/2019/Conference_

### Official Review · AnonReviewer2 · 2018-11-03
**Potentially Interesting Method but the Paper Needs Polishing**

**Rating:** 3
**Confidence:** 4

**Review:**

The paper presents a new Generative Adversarial Network (GAN) for learning a
target distribution that is defined as the difference between two other
distributions. Applications in semi-supervised learning and adversarial training
are considered in the experimental evaluation and results are presented in
computer vision tasks.

The paper is not very well written and can be hard to follow. One very important
issue for me was motivation for defining the target distribution as a difference
between two other distributions. I am not familiar with this area, but reading
through the introduction it was never clear to me why this is a useful scenario,
in practice. Furthermore, some statements in the introduction felt quite
arbitrary. For example, the authors state that PixelCNN "does not have a latent
representation" in a manner that makes it sound as if that is a bad thing. If
indeed it is, then why so? It would be very helpful to motivate the setting more
and to provide a couple of examples of where this method would be useful, in the
introduction. Also, regarding the MNIST example in the end of page 1, what is
the "universal set"? This paragraph also felt a bit arbitrary and unclear.

Some comments about the rest of the paper:
  - The theoretical results of section 3 are just stated/listed, but are not
    connected to algorithm 1. Please connect them to the different parts of the
    algorithm and state in a couple sentences what they imply for the algorithm.
  - Right after theorem 1, which assumption are you referring to when you say
    "the assumption in Theorem 1"?
  - The reformulation of section 3.1 is never justified. What led you to use
    this reformulation and why do you think it is more stable in practice?
  - You should mention in the caption of table 4, what quantity you are
    computing.

Note that my evaluation for this paper is based mainly on the way it is written
as, in its current state, it is hard for me to judge what is novel and what is
useful, and what readers are supposed to take in by reading this paper. The main
question that the paper definitely needs to answer, but does not do so currently
(in my opinion) is:

  When is this method useful to readers? For solving which problems and under
  what conditions? And also, when is this method bad and should not be used?

== Experiments ==

Section 5.1 is hard to follow and I don't quite get how it connects to the rest.

Also, in section 5.1.2 you mention that in comparison to Dai et al. (2017) your
method does not need to rely on an additional density estimation network. Even
if that is true, I cannot see how it is a useful remark given that the method of
Dai et al. seems to always beat your method.

== Style ==

In figure 1, no labels or legends are provided making it hard to figure out
what's going on at a glance. It would be very helpful to include labels and a
legend.

Equation 2 is not written correctly. The equals sign only refers to "V(G, D)"
and not the min-max of that, right? Please make that explicit by first defining
"V(G, D)" alone.

---

> ### Author Response · Authors · 2018-11-19
> **Thanks for your review.**
>
> We revise the manuscript to answer the comments.
>
>
> >> “Some statements in the introduction felt quite arbitrary. For example, the authors state that PixelCNN "does not have a latent representation" in a manner that makes it sound as if that is a bad thing. If indeed it is, then why so? It would be very helpful to motivate the setting more and to provide a couple of examples of where this method would be useful, in the introduction.”
> >> “When is this method useful to readers? For solving which problems and under what conditions? And also, when is this method bad and should not be used?”
>
> We rephrase the first paragraph in original paragraph to delete some discussions which may be out-of-scope such as PixelCNN and VAE in this paper. We also add a subsection Sec. 1.1 to emphasize our motivations along with the scenario, where the proposed DSGAN will have the ability to generate diverse unseen data that is helpful for one-class classification, semi-supervised learning, and adversarial attacks. Those techniques are important and involve many real applications shown in Fig. 8 in the revised manuscript.
>
>
> >> “The theoretical results of section 3 are just stated/listed, but are not connected to algorithm 1. Please connect them to the different parts of the algorithm and state in a couple sentences what they imply for the algorithm.”
>
> The theoretical results in Sec. 3 aims to explain why the objective function in Eq. (2) can learn the wanted distribution. But, Algorithm 1 is to explain the training procedure which may be not directly related to theoretical results.
>
>
> >>”You should mention in the caption of table 4, what quantity you are computing”
>
> We replace table 4 (in the original manuscript) with Fig. 7 (in the revised manuscript) for the comprehensive evaluation on both robustness and performance of the models. And the quantity we computed as x-axis (same as the quantity in original table 4) is “epsilon”, which is the l_2 (or l_inf) norm between original image and corresponding adversarial example. The description of epsilon is also added in the caption of Fig. 7.
>
>
> >> “Section 5.1 is hard to follow and I don't quite get how it connects to the rest.”
>
> We rephrase the descriptions of Sec 5.1 and hope it is more understandable. Briefly saying, Dai et al. (2017) claim that if the generator can satisfy the conditions Eq. (8) and Eq. (9), it can help the semi-supervised learning through the objective function Eq. (5). In our method, if we set the p_{\bar{d}} to the linear combination of labeled and unlabeled data, then the generator can learn the distribution which satisfies both Eq. (8) and Eq. (9). We use this generator to help semi-supervised learning further.
>
>
> >> “Also, in section 5.1.2 you mention that in comparison to Dai et al. (2017) your method does not need to rely on an additional density estimation network. Even if that is true, I cannot see how it is a useful remark given that the method of Dai et al. seems to always beat your method.”
>
> For comparison with Dai et al. (2017) in Sec. 5.1.2, additional density estimation network may not exist always and be expensive to train (also discussed in Lee et al. (2018)). In the contrast, DSGAN is a simple and effective way for generating complementary data.
> We also claim that the generator might not perfectly learn the distribution we assigned, however, this problem will be attenuated since there are a number of researches on solving this problem.  In Dai et al. (2017), their method relies on feature matching, which aim to match the first-order statistics of the distribution, and it cannot take all the advantages of the progress of GANs. Due to this reason, our method has more benefits in the long term.  The discussion in detail is also in Sec. 5.1.2.
>
>
> >> “In figure 1, no labels or legends are provided making it hard to figure out what's going on at a glance. It would be very helpful to include labels and a legend.”
>
> We add the legends to those figures and put more illustrations for our idea. Hope those are helpful for understanding our methods.
>
>
> >>The reformulation of section 3.1 is never justified. What led you to use this reformulation and why do you think it is more stable in practice?
>
> From the view of theory, the reformulation is equivalent to original formula. The original reason is that we can use 2m samples instead of 3m samples for each mini-batch. However, we found that the performance also becomes better. Thus, we conjecture that the equivalence may be based on the linearity of expectation, but mini-batch stochastic gradient descent in practical training may lead to the different outcomes. Due to the limit of space, we move Sec. 3.1 (in the original manuscript) to Appendix C (in the revised manuscript).
>
>
> >>Right after theorem 1, which assumption are you referring to when you say "the assumption in Theorem 1"?
>
> We rephrase this sentence “the assumption in Theorem 1” as "The assumption, α p_{d}(x) ≤ p_{\bar{d}}(x) for all x’s in Theorem 1".

---

### Official Review · AnonReviewer3 · 2018-11-05
**An interesting problem formulation, but the method doesn't seem innovative and the empirical results not very convincing**

**Rating:** 4
**Confidence:** 3

**Review:**

- Summary:
This paper considers the problem of learning a GAN to capture a target distribution p_t with only very few training samples from p_t available.

- Good
An interesting problem formulation.
The proposed approach is not new, but seems to be a sensible and simple solution to the problem formulated in this paper. I would see the contributions of the paper: (1) an interesting problem formulation on how to learn p_t (with a few assumptions) (2) a sensible adaptation of GANs on this problem (with minor modifications to GANs which have been observed/adopted in many GAN literatures in the last two years)
The training appaoches/tricks are rather straightforward and not new as well.

- Suggestions
The main problem of this paper is that it does not provide sufficient results on any real applications that can support its problem&model formulations.
For example, in which scenarios would the users of the model need to train a GAN to mimic a target distribution p_t which is a difference of another two distributions (with examples available there but unavailable in p_t)? It would be good to show significant results on real applications to show the problem and the method useful.

Two applications on semi-supervised classification and adversarial training are discussed. While both seem to be very artificial IMO if considering the used dataset and designed experiments. The results are also not convincing even for the shown two experiments compared to baselines.

No related works on addressing the similar problems have been discussed nor compared in experiments.

- Theoretical results:
While the authors claim new theoretical results, in fact, I didn't see any contributions here as the theories developed in section 3 are mostly rather straightforward. There have been some similar theories being developed in previous papers where a component in GAN exhibits mixture-modeled forms, such as in TripleGan (Li et al. NIPS'17). So I would not recommend the authors to claim contributions here.

- Writing:
The paper does not seem to be polished. It may not be necessary to exceed 8 pages as many spaces in this paper could be easily squeezed (apparently). The organization could be better; Some parts are vague and difficult to understand;  the writing could be improved to be more clearly demonstrate the contributions of this paper.

---

> ### Author Response · Authors · 2018-11-19
> **Thanks for your review.**
>
> We revise the manuscript to answer the comments.
>
>
> >> “The main problem of this paper is that it does not provide sufficient results on any real applications that can support its problem&model formulations. For example, in which scenarios would the users of the model need to train a GAN to mimic a target distribution p_t which is a difference of another two distributions (with examples available there but unavailable in p_t)? It would be good to show significant results on real applications to show the problem and the method useful.”
> “Two applications on semi-supervised classification and adversarial training are discussed. While both seem to be very artificial IMO if considering the used dataset and designed experiments. The results are also not convincing even for the shown two experiments compared to baselines.”
>
> For responding the suggestions, we add a subsection Sec. 1.1 to emphasize our motivations. Our goal is to generate unseen data, which is absent during training process and is difficult to collect. We show that the proposed DSGAN will have the ability to generate diverse unseen data that is helpful for one-class classification, semi-supervised learning, and adversarial attacks. Those techniques are important and involve many real applications shown in Fig. 8 in the revised manuscript.
>
>
> >>”No related works on addressing the similar problems have been discussed nor compared in experiments.”
>
> We also add Sec. 6 in the revised manuscript to discuss related works about generating unseen data. In sum, DSGAN is a simple and effective way for our objective. The results compared with other methods are discussed in Sec. 5.1.2 and 5.2.2 corresponding to the two experiments.
>
>
> >> “While the authors claim new theoretical results, in fact, I didn't see any contributions here as the theories developed in section 3 are mostly rather straightforward. There have been some similar theories being developed in previous papers where a component in GAN exhibits mixture-modeled forms, such as in TripleGan (Li et al. NIPS'17). So I would not recommend the authors to claim contributions here.”
>
> About the contribution of theoretical proofs (in Sec. 1.2 in the revised manuscript), we remove the sentences “when the optimum of the Jensen-Shannon divergence is attained such that the generator distribution is equal to the target distribution,” which has been covered under [Goodfellow et al. 14]. Instead we show that the proposed DSGAN can learn the distribution which support set is difference of support sets of two distributions shown in Proposition 2.
>
>
> >> The paper does not seem to be polished. It may not be necessary to exceed 8 pages as many spaces in this paper could be easily squeezed (apparently). The organization could be better; Some parts are vague and difficult to understand; the writing could be improved to be more clearly demonstrate the contributions of this paper.
>
> Consequently, in the revised manuscript, we reorganize some sections and rephrase some sentences for better understandings.

---

### Official Review · AnonReviewer1 · 2018-11-12
**Interesting but straightforward idea that needs more development.**

**Rating:** 5
**Confidence:** 4

**Review:**

The paper presents DS-GAN, which aims to learn the difference between any two distributions whose samples are difficult or impossible to collect. To this end they simply model the target distribution such that adding it to one of the distribution results in another, and propose a min-max objective based on it. To show the effectiveness of the proposed DS-GAN, the authors validate it on semi-supervised learning and adversarial training tasks, on which it performs reasonably well in generating the difference between the two distributions.

Pros
- The idea of learning the difference between two distributions is novel to my knowledge. Similar ideas have been explored in prior work such as [Li et al. 17] but are not doing exactly what the authors try to do.
- The proposed method works reasonably well on semi-supervised learning and adversarial learning tasks, and thus it seems practically useful.

Cons
- The proposed model is quite straightforward in its formulation, and since the paper is not addressing the importance of, or any challenges with the problem they are trying to solve, the contribution of this work appears minor.
- The authors list theoretical results as contributions, but they are rather straightforward replacement of the p_d and p_g terms in the original theorems on optimality in [Goodfellow et al. 14] with the target distributions in this paper, that has nothing to do with what the authors claim in this paper.  Thus they add nothing to the value of the paper.
- The motivation is very unclear when reading the introduction section, and Figure 1 does not do a good job of providing it.
- The experimental validation is lacking in many aspects. I think the main results should show that the difference-seeking GAN can learn distributional differences but the authors jump straight to the applications. Also, the current experimental section simply reports performances on the two tasks, without much analysis showing why it works well and how it works differently from other models.

In sum, the idea seems nice and interesting but the model is straightforward and the current results are very weak in analysis in order to make a good paper. I would recommend the authors to perform further analysis of the model either theoretically or experimentally.

---

> ### Author Response · Authors · 2018-11-19
> **Thanks for your review.**
>
> We revise the manuscript to answer the comments.
>
> >> “The proposed model is quite straightforward in its formulation, and since the paper is not addressing the importance of, or any challenges with the problem they are trying to solve, the contribution of this work appears minor.”
> >> “The motivation is very unclear when reading the introduction section, and Figure 1 does not do a good job of providing it. “
>
> We add a subsection Sec. 1.1 to emphasize our motivations. Our goal is to generate unseen data, which is absent during training process and is difficult to collect. We show that the proposed DSGAN will have the ability to generate diverse unseen data that is helpful for one-class classification, semi-supervised learning, and adversarial attacks. Those techniques are important and involve many real applications shown in Fig. 8 in the revised manuscript.
>
>
> >> “The authors list theoretical results as contributions, but they are rather straightforward replacement of the p_d and p_g terms in the original theorems on optimality in [Goodfellow et al. 14] with the target distributions in this paper, that has nothing to do with what the authors claim in this paper. Thus they add nothing to the value of the paper.”
>
> We rephrase the contribution about theoretical proofs (in Sec. 1.2 in the revised manuscript).  We remove the sentences “when the optimum of the Jensen-Shannon divergence is attained such that the generator distribution is equal to the target distribution,” which has been covered under [Goodfellow et al. 14]. Instead we show that the proposed DSGAN can learn the distribution which support set is difference of support sets of two distributions shown in Proposition 2.
>
>
> >> “The experimental validation is lacking in many aspects. I think the main results should show that the difference-seeking GAN can learn distributional differences but the authors jump straight to the applications. Also, the current experimental section simply reports performances on the two tasks, without much analysis showing why it works well and how it works differently from other models. “
>
> We rearrange the figures, where Figs. 2-6 in the revised manuscript illustrate that DSGAN learn distributional difference on both toy-data and MNIST. We also add more discussions to compare our method with other methods in Sec 5.1.2 and Sec 5.2.2, and emphasize the advantages in DSGAN in the last paragraph of Sec. 5.1.2.

---

### Author Response · Authors · 2018-11-19
**We upload the revised manuscript .**

Thanks for all the comments.

We list all the modifications as following:

1. We add Sec. 1.1 to describe our motivation and scenario.

2. We reorganize Sec. 2.2 to clarify the main idea of DSGAN through several case studies.

3. We move Sec. 3.1 in the original manuscript to appendix C in the new one.

4. We rephrase Sec. 5.1 for better understanding. Moreover, we replace Tables 4 and 5 in the original manuscript with Figs. 7, 9 and 10 in the new one for more comprehensive evaluation.

5. We have more discussion to our model in Sec. 5.1.2 and Sec. 5.2.2.

6. We add Sec. 6 to discuss related works.

---

### Meta-Review · Area_Chair1 · 2018-12-12
**Lack of novelty**

**Confidence:** 4
**Recommendation:** Reject

**Metareview:**

The paper presents a GAN for learning a target distribution that is defined as the difference between two other distributions.

The reviewers and AC note the critical limitation of novelty and appealing results of this paper to meet the high standard of ICLR.

AC thinks the proposed method has potential and is interesting, but decided that the authors need more works to publish.